# Hematological Biomarkers for Early Detection of Lung Cancer: Evaluating the Diagnostic Potential of Circulating Interleukin Levels

**DOI:** 10.3390/ijms262211014

**Published:** 2025-11-14

**Authors:** Ouafaa Morjani, Yi-Wei Yang, Rachid Lahlil, Hamid Lakhiari, Hassan Alaoui

**Affiliations:** 1Laboratory of Physical Chemistry and Biotechnology of Biomolecules and Materials, Faculty of Sciences and Techniques of Mohammedia, Hassan II University of Casablanca, Mohammedia 28800, Morocco; morjaniouafaa@gmail.com (O.M.); lakhiari@gmail.com (H.L.); 2Department of Surgery, Thoracic Oncology Program, University of California, San Francisco, CA 94143, USA; yiwei.yang@ucsf.edu (Y.-W.Y.);; 3Institute for Research in Hematology and Transplantation (IRHT), Hasenrain Hospital, 87 Avenue d’Altkirch, 68100 Mulhouse, France

**Keywords:** interleukin biomarkers, lung cancer, NSCLC, early detection, inflammatory cytokines

## Abstract

Early detection of lung cancer remains a major unmet clinical need, as most patients are diagnosed at advanced stages when curative treatment options are limited. Circulating cytokines and interleukins represent promising molecular biomarkers for the non-invasive diagnosis and monitoring of tumor development. In this study, we investigated the diagnostic potential of plasma interleukins in distinguishing early-stage non-small cell lung cancer (NSCLC) from healthy individuals and patients with chronic obstructive pulmonary disease (COPD). Quantitative analyses demonstrated significantly elevated plasma levels of IL-1RA, IL-6, IL-8, IL-10, and IL-17A in NSCLC patients compared with healthy controls. Among these, a composite biomarker panel comprising IL-6, IL-10, IL-8, and IL-1RA exhibited the highest diagnostic performance, outperforming individual cytokines and other combinations. This interleukin-based signature also differentiated NSCLC from COPD with strong specificity, underscoring its potential clinical applicability. These findings highlight the molecular and translational relevance of plasma interleukin profiling as a non-invasive diagnostic approach for early lung cancer detection, potentially enabling earlier intervention and improved patient outcomes.

## 1. Introduction

Lung cancer remains a major global health burden and the foremost cause of cancer-related mortality worldwide, reflecting both its aggressive biological nature and the frequent delay in diagnosis [1]. Non-small cell lung cancer (NSCLC) constitutes approximately 80–85% of all lung cancer cases, whereas small cell lung cancer (SCLC) accounts for 10–15%. According to the 2021 World Health Organization (WHO) classification of thoracic tumors, NSCLC primarily includes adenocarcinoma, squamous cell carcinoma, and large-cell carcinoma—adenocarcinoma being the most prevalent subtype. This classification provides a unified histopathological framework that facilitates clinical diagnosis and highlights the potential relevance of cytokine-based biomarkers across diverse NSCLC subtypes.

Globally, lung cancer was responsible for an estimated 20 million new cases and 1.8 million deaths in 2022, representing nearly one in five cancer-related fatalities [2]. Projections suggest that the global cancer burden will rise by more than 70% by 2050, posing a significant challenge to healthcare systems worldwide. In the United States, approximately 234,580 new lung cancer cases and 125,070 deaths are projected in 2024 [3,4,5]. Despite advances in surgery, radiotherapy, and systemic therapies, the prognosis for lung cancer remains poor, primarily due to late-stage diagnosis. While surgical resection offers curative potential for early-stage disease, most patients present with advanced tumors, resulting in a dramatic decline in survival rates—from 73% in localized disease to only 18% in advanced stages [6,7,8]. These data underscore the critical need for improved strategies enabling early detection, accurate staging, and effective non-invasive monitoring [9].

Recent advances in liquid biopsy technologies offer new opportunities for early cancer detection and disease monitoring. Circulating tumor DNA (ctDNA), tumor-derived exosomes, and cell-free nucleic acids have emerged as promising sources of diagnostic and prognostic biomarkers [10,11,12]. These analytes enable minimally invasive tumor detection, assessment of therapeutic response, identification of minimal residual disease, and early relapse prediction. Among such approaches, epigenetic profiling, particularly DNA methylation analysis, demonstrates superior sensitivity compared with mutation-based methods [13]. This advantage stems from the high prevalence and tissue specificity of aberrant methylation patterns across tumors, which allow accurate identification of tumor origin despite genetic heterogeneity. However, translation of these technologies into routine clinical practice remains challenging due to the absence of standardized protocols for sample preparation, exosome isolation, and methylation analysis, as well as the technical complexity of CpG conversion workflows [14]. Addressing these methodological barriers is essential to realize the full clinical potential of blood-based molecular diagnostics.

Beyond nucleic acid biomarkers, circulating interleukins have attracted increasing interest as immunologically informative and clinically accessible biomarkers. Pro-inflammatory cytokines such as IL-6 and IL-8 are frequently elevated in lung cancer and contribute to tumor progression by promoting angiogenesis, proliferation, and immune evasion [15,16]. Conversely, dysregulation of anti-inflammatory cytokines such as IL-10 impairs immune surveillance and facilitates tumor growth [17]. The balance between pro- and anti-inflammatory cytokines therefore reflects the dynamic interaction between tumor cells and the host immune system. Monitoring these interleukins offers a minimally invasive, cost-effective approach for early detection and disease monitoring, potentially complementing imaging modalities such as low-dose computed tomography (LDCT).

In this study, we investigated the diagnostic utility of a plasma interleukin panel comprising IL-6, IL-10, IL-8, and IL-1RA to distinguish early-stage NSCLC from healthy individuals and patients with chronic obstructive pulmonary disease (COPD). This composite panel demonstrated strong discriminatory power, underscoring the potential of interleukin profiling as a robust and clinically feasible approach for early NSCLC detection. Furthermore, elucidating cytokine signatures associated with tumor immune modulation may deepen our understanding of NSCLC immunobiology—particularly in relation to immune checkpoint pathways such as PD-L1 and CTLA-4—and contribute to the development of integrated diagnostic and therapeutic strategies.

## 2. Results

### 2.1. Circulating Signaling Protein Blood Levels Are Increased in a Preclinical Human NSCLC Xenograft Model

To investigate circulating factors associated with tumor burden, we quantified plasma analytes released by tumor cells in a preclinical human non-small cell lung cancer (NSCLC) xenograft model. Plasma samples from mice bearing xenografts derived from the H292 NSCLC cell line were analyzed using a validated array-based multiplex sandwich ELISA system (Eve Technologies), encompassing 32 signaling proteins.

Compared with non-tumor-bearing controls, xenograft-bearing mice exhibited a marked elevation in circulating levels of several analytes, including IL-6, IL-8, IL-10, IL-2, growth-regulated oncogene (GRO), granulocyte colony-stimulating factor (G-CSF), macrophage-derived chemokine (MDC), and vascular endothelial growth factor (VEGF) (Figure 1). Although inter-animal variability in cytokine concentrations was noted, this variation reflects biological heterogeneity and the intrinsic dynamic range of cytokine production within tumor-bearing hosts. Importantly, despite individual differences, the overall pattern of cytokine elevation remained consistent across all xenograft-bearing mice, underscoring the robustness of tumor-associated systemic cytokine modulation in this NSCLC model.

### 2.2. Interleukin Plasma Levels Are Significantly Elevated in Early-Stage NSCLC Patients

To evaluate the clinical relevance of circulating cytokines identified in the xenograft model, we next analyzed plasma interleukin concentrations in patients with early-stage non-small cell lung cancer (NSCLC) compared with healthy individuals and subjects with chronic obstructive pulmonary disease (COPD). The study cohort included 40 participants: 22 patients with histologically confirmed NSCLC, 6 patients with COPD, and 12 age- and sex-matched healthy controls. The clinical characteristics of the participants are summarized in Table 1. None of the NSCLC patients had concurrent chronic inflammatory conditions. Among the NSCLC group, 11 cases were classified as stage IA, 7 as stage IB, and 4 as stage IIA, with a mean age of approximately 60 years (range 60–84).

Sub-analysis by disease stage revealed a trend toward higher plasma IL-6 and IL-10 concentrations in stage IIA compared with stages IA and IB, although these differences did not reach statistical significance (*p* > 0.05). This pattern nonetheless suggests that cytokine levels may increase with disease progression in early NSCLC. When compared with healthy controls, NSCLC patients exhibited significantly elevated plasma levels of multiple inflammatory mediators, including IL-6, IL-8, IL-10, IL-17A, and IL-1RA (Figure 2). To control for the confounding influence of chronic inflammation, cytokine levels were also compared with those in COPD patients. NSCLC plasma samples demonstrated markedly higher concentrations of IL-1RA, IL-6, IL-8, IL-10, and IL-17A than COPD samples, supporting a potential association between malignant transformation and heightened systemic interleukin activity.

Notably, the elevation of IL-6, IL-8, and IL-10 observed in patients paralleled that seen in the H292 xenograft model, suggesting these cytokines may represent conserved signatures of NSCLC-associated systemic inflammation. Conversely, differences in other cytokines between the clinical and preclinical datasets may reflect species-specific immune responses or differences in tumor burden and microenvironmental context. The local inflammatory milieu—characterized by lymphocytic, macrophagic, or neutrophilic infiltration—likely influences interleukin production. However, due to the retrospective design and limited availability of matched tissue specimens, systematic grading of tumor-associated inflammation could not be performed.

Future studies should incorporate immunohistochemical analyses of tumor-infiltrating immune cells (e.g., CD4^+^ and CD8^+^ T cells, IL-6R expression, macrophage markers) to delineate the relationship between systemic cytokine alterations and intratumoral immune activity. Collectively, these findings highlight that plasma interleukin profiling may serve as a promising non-invasive approach for distinguishing early-stage NSCLC from both healthy and inflammatory lung conditions, warranting further validation in larger prospective cohorts.

### 2.3. Comparable Plasma Levels of IL-1A, IL-6, IL-8, IL-10, and IL-17A Between Adenocarcinomas and Squamous Cell Carcinoma, the 2 Major NSCLC Pathological Subtypes of NSCLC

To determine whether circulating interleukin profiles differ between the major pathological subtypes of non-small cell lung cancer (NSCLC), plasma concentrations of IL-1RA, IL-6, IL-8, IL-10, and IL-17A were compared between patients with adenocarcinoma and those with squamous cell carcinoma. Statistical analysis was performed using the Kruskal–Wallis test, as summarized in Table 2.

Although the adenocarcinoma group exhibited higher mean IL-1RA levels (128.00 ± 62.04 pg/mL) compared to the squamous cell carcinoma group (38.35 ± 10.41 pg/mL), this difference did not reach statistical significance (*p* = 0.20). Likewise, no significant intergroup differences were detected for IL-6 (*p* = 0.41), IL-8 (*p* = 0.92), IL-10 (*p* = 0.72), or IL-17A (*p* = 0.40). These findings suggest that circulating interleukin concentrations are not substantially influenced by NSCLC histological subtype. Consequently, subsequent analyses and biomarker validation efforts may consider NSCLC as a unified disease category for cytokine-based biomarker evaluation.

### 2.4. Diagnostic Efficacy of IL-1A, IL-6, IL-8, IL-10, and IL-17A Expression in Lung Cancer

To assess the diagnostic potential of circulating interleukins as biomarkers for NSCLC detection, logistic regression and receiver operating characteristic (ROC) curve analyses were performed for each cytokine individually and in selected combinations. The diagnostic performance of each marker was quantified using the area under the ROC curve (AUC) as an index of discriminative power (Figure 3 and Figure 4).

Among individual cytokines, IL-6 (AUC = 0.974), IL-8 (AUC = 0.970), and IL-10 (AUC = 0.924) demonstrated superior diagnostic accuracy compared to IL-1RA (AUC = 0.688) and IL-17A (AUC = 0.731). The 95% confidence intervals for IL-6, IL-8, and IL-10 were 0.9277–1.000, 0.9220–1.000, and 0.8376–1.000, respectively, whereas IL-1RA (0.5061–0.8709) and IL-17A (0.5625–0.8996) exhibited broader and lower ranges. These findings identify IL-6, IL-8, and IL-10 as the most robust individual diagnostic biomarkers for distinguishing NSCLC patients from non-cancer controls.

Importantly, multiplex analysis of combined cytokine panels further enhanced diagnostic discrimination compared to single-analyte evaluation. The composite panel consisting of IL-6 + IL-10 + IL-8 + IL-1RA achieved the highest AUC value (0.996), outperforming all individual markers and other tested combinations. Panels comprising IL-6 + IL-10 + IL-8 + IL-17A + IL-1RA and IL-6 + IL-10 + IL-8 + IL-17A yielded comparably high AUCs (0.992), underscoring the additive diagnostic value of multiplex cytokine profiling.

Collectively, these results indicate that a multi-analyte interleukin panel, particularly IL-6 + IL-10 + IL-8 + IL-1RA, markedly improves diagnostic accuracy for NSCLC. This suggests a promising role for combined cytokine profiling as a non-invasive, blood-based diagnostic approach for early detection and clinical monitoring of lung cancer.

## 3. Discussion

Lung cancer remains a major global health challenge, primarily due to its asymptomatic onset and frequent diagnosis at advanced stages, which significantly limits therapeutic options and contributes to poor survival outcomes [17,18,19]. The present study underscores the critical importance of early detection, emphasizing that timely identification of tumor-associated systemic changes can markedly improve prognosis and patient survival [18,19]. Among emerging molecular indicators, circulating cytokines—particularly interleukins—play a central role in orchestrating immune and inflammatory responses and have shown promise as non-invasive biomarkers for early cancer detection [20,21,22,23,24].

Using validated multiplex ELISA assays (Eve Technologies platform), we profiled a broad range of circulating signaling proteins in both a preclinical NSCLC xenograft model and in human patient cohorts. Our analyses revealed a consistent increase in multiple inflammatory interleukins—specifically IL-1RA, IL-6, IL-8, IL-10, and IL-17A—in the plasma of early-stage NSCLC patients and xenograft-bearing mice compared with controls. These findings support the concept that tumor presence induces a characteristic systemic cytokine response, which may serve as a surrogate indicator of early malignant transformation. Elevated levels of these cytokines in NSCLC patients, as compared to both healthy subjects and individuals with COPD, further suggest that cancer-related inflammation is distinct from non-neoplastic chronic inflammatory processes.

Previous studies have highlighted IL-1β, IL-6, and IL-8 as key mediators of tumor-associated inflammation, with IL-1β expression correlating with disease progression and survival in molecularly defined NSCLC subgroups [25]. Similarly, circulating chemokines such as CCL11, CCL2, and CCL13 have been implicated as complementary biomarkers for NSCLC detection and stratification [26]. Our results expand on these findings by providing comparative evidence that several interleukins are concurrently upregulated in early-stage NSCLC and that a composite interleukin panel yields superior diagnostic accuracy compared with individual markers.

Non-small cell lung cancer (NSCLC), which accounts for approximately 85% of all lung cancers, continues to pose significant diagnostic and therapeutic challenges. Despite substantial advances in molecular diagnostics [27,28] and targeted therapies, mortality remains high, largely due to late-stage detection and tumor heterogeneity. Accumulating evidence implicates chronic inflammation as a pivotal contributor to oncogenesis, tumor progression, and immune evasion [29,30]. Pro-inflammatory interleukins such as IL-6, IL-8, IL-10, IL-17A, and IL-1RA regulate critical biological pathways including angiogenesis, proliferation, invasion, and immune modulation [31,32,33,34]. IL-6 and IL-8, in particular, have been linked to tumor growth and neovascularization, and their overexpression has been correlated with poor clinical outcomes in NSCLC [23,35,36,37,38]. Conversely, IL-10 plays a dual role—suppressing excessive inflammation while contributing to tumor immune escape through inhibition of cytotoxic T-cell activity [39,40,41].

In the present study, plasma concentrations of IL-1RA, IL-17A, IL-6, IL-8, and IL-10 were significantly higher in NSCLC patients compared to healthy controls, and markedly elevated relative to COPD patients, highlighting their specificity to malignancy-associated inflammation. Although IL-8 levels were also increased in COPD, likely reflecting its role in chronic airway inflammation, the overall cytokine profile of NSCLC patients remained distinct. Importantly, no significant differences in cytokine concentrations were detected between adenocarcinoma and squamous cell carcinoma subtypes, suggesting that these biomarkers reflect core NSCLC pathophysiology rather than subtype-specific biology.

Diagnostic modeling further confirmed the strong discriminative performance of IL-6, IL-8, and IL-10, individually and in combination. The multiplex panel comprising IL-6 + IL-10 + IL-8 + IL-1RA achieved the highest diagnostic accuracy (AUC = 0.996), outperforming single markers and alternative combinations. These findings support the clinical utility of multi-analyte cytokine profiling as a sensitive and specific blood-based strategy for early NSCLC detection.

Emerging data also link elevated IL-6 and IL-10 levels with increased PD-L1 expression and reduced cytotoxic T-cell activity in NSCLC, suggesting an intersection between inflammatory signaling and immune checkpoint regulation [17,25]. Integrating cytokine signatures with established immunomodulatory biomarkers such as PD-L1 and CTLA-4 may enhance patient stratification and predictive modeling for immunotherapy responsiveness. Although checkpoint expression was not assessed in our current cohort, its evaluation in forthcoming prospective studies may provide deeper mechanistic insight into cytokine-driven immune regulation.

### Study Limitations

We acknowledge that the relatively small sample size of our clinical cohort represents a limitation of the present study and may introduce potential statistical bias. Although the observed cytokine trends were consistent and biologically meaningful, the limited number of cases restricts statistical power and generalizability. This pilot-scale analysis should therefore be regarded as exploratory, generating hypotheses. Future studies involving larger, prospectively enrolled, and multi-center cohorts are warranted to validate these results and further establish the clinical utility of plasma interleukin profiling for early NSCLC detection.

In summary, our findings demonstrate that circulating interleukins—particularly IL-1RA, IL-6, IL-8, IL-10, and IL-17A—hold substantial promise as early diagnostic biomarkers for NSCLC. The observed consistency across preclinical and clinical settings reinforces their translational potential. Profiling such cytokines from peripheral blood offers a minimally invasive approach that could complement imaging and molecular diagnostics, improving early detection and personalized treatment planning.

## 4. Materials and Methods

### 4.1. Plasma Samples

Human plasma specimens were obtained from the Thoracic Tissue Bank of the Thoracic Oncology Laboratory at the University of California, San Francisco (UCSF), California, USA. The study cohort included treatment-naïve patients newly diagnosed with histologically confirmed, early-stage non-small cell lung carcinoma (NSCLC). All blood samples were collected prior to the initiation of any systemic therapy or surgical resection to minimize confounding effects on circulating cytokine levels. All patients subsequently underwent surgical treatment at UCSF. Plasma samples from healthy donors were obtained commercially from Vital Products, Inc. (Boynton Beach, FL, USA).

### 4.2. Xenograft Mouse Models for Plasma Collection

Human NSCLC xenograft models were established using H292 cells cultured in RPMI-1640 medium supplemented with 10% fetal bovine serum (FBS). Under the supervision of the UCSF Preclinical Therapeutics Core, a total of 10^7^ H292 cells were implanted subcutaneously into female athymic Nu/Nu mice aged 7–9 weeks. Tumor growth was monitored until the mean volume reached approximately 600 mm^3^, at which point peripheral blood samples were collected into EDTA-coated tubes. Plasma was separated by centrifugation and stored at −80 °C until analysis.

All xenografts were generated in immunodeficient female Nu/Nu mice lacking mature T cells, a model that allows stable engraftment of human NSCLC cells while precluding full adaptive immune responses. Consequently, although this system enables evaluation of tumor-derived circulating factors, it does not fully capture the complexity of host–tumor immune interactions.

### 4.3. Multiplex Cytokine Quantification

Quantitative assessment of circulating cytokines was performed using a validated array-based multiplex sandwich ELISA platform (Eve Technologies, Calgary, AB, Canada). This system enables simultaneous measurement of 33 human analytes with high sensitivity and reproducibility. Plasma samples from age- and sex-matched healthy individuals (n = 12), patients with chronic obstructive pulmonary disease (COPD; n = 6), and early-stage NSCLC patients (n = 22) were analyzed in parallel. The multiplex array provided comprehensive comparative profiles of circulating signaling molecules across all study groups.

### 4.4. Statistical Analysis

All statistical analyses were performed using SPSS Statistics version 25.0 (IBM Corp., Armonk, NY, USA), and graphical data visualization was conducted with GraphPad Prism version 10.0 (GraphPad Software, San Diego, CA, USA). Data were assessed for normality, and non-normally distributed variables are presented as medians with interquartile ranges (P25–P75). Comparisons of analyte concentrations among study groups and between NSCLC pathological subtypes were conducted using the Kruskal–Wallis test. Differences between two independent groups were further evaluated using the *t*-test with Welch’s correction for unequal variances. To assess the diagnostic value of each cytokine, logistic regression analysis was performed to estimate predictive probabilities, followed by generation of receiver operating characteristic (ROC) curves. Diagnostic accuracy was quantified by the area under the ROC curve (AUC), where AUC < 0.5 indicated poor discrimination and values approaching 1 denoted excellent diagnostic performance. Statistical significance was defined as *p* < 0.05.

## 5. Conclusions

This study identifies circulating interleukins—particularly IL-1RA, IL-6, IL-8, IL-10, and IL-17A—as promising blood-based biomarkers for the early detection of non-small cell lung cancer (NSCLC). Plasma levels of these cytokines were significantly higher in NSCLC patients than in healthy individuals or those with chronic obstructive pulmonary disease, highlighting their specificity for malignancy-associated inflammation.

A composite panel combining IL-6, IL-8, IL-10, and IL-1RA achieved superior diagnostic accuracy compared with individual markers, underscoring the value of multiplex cytokine profiling as a minimally invasive diagnostic tool. The concordance between clinical and preclinical data further supports the translational relevance of these interleukins in NSCLC pathophysiology.

While encouraging, these findings should be interpreted in light of the limited cohort size, which warrants validation in larger, independent, and prospectively designed studies. Future work should also explore integration with molecular and immunologic biomarkers such as PD-L1 and circulating tumor DNA to enhance diagnostic precision and support the development of non-invasive, personalized screening approaches for early NSCLC detection.

## Figures and Tables

**Figure 1 ijms-26-11014-f001:**
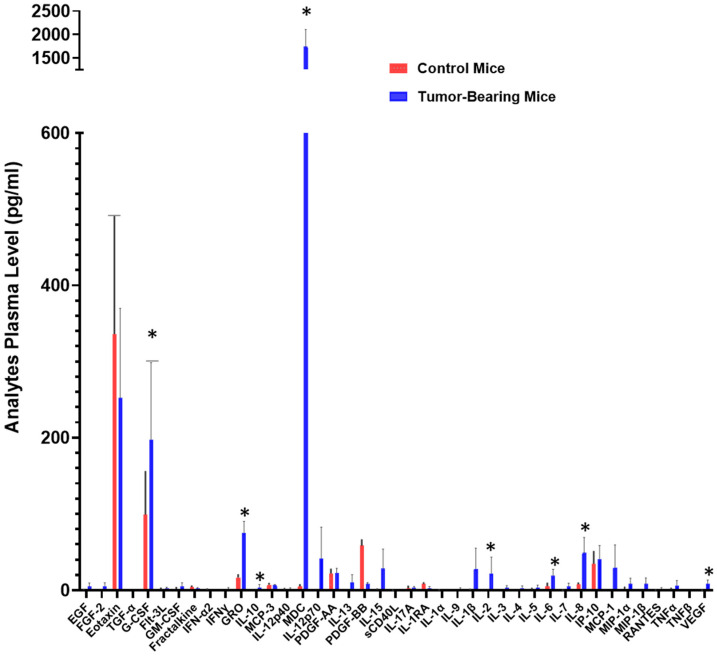
Cytokine levels potentially mobilized into the bloodstream. Plasma samples from three mice bearing xenograft tumors derived from the H292 NSCLC cell line and three control mice were analyzed using a commercially available and validated array-based multiplex sandwich ELISA system (Eve Technologies), targeting 32 analytes. (*) denotes a *p*-value ≤ 0.05, indicating statistical significance.

**Figure 2 ijms-26-11014-f002:**
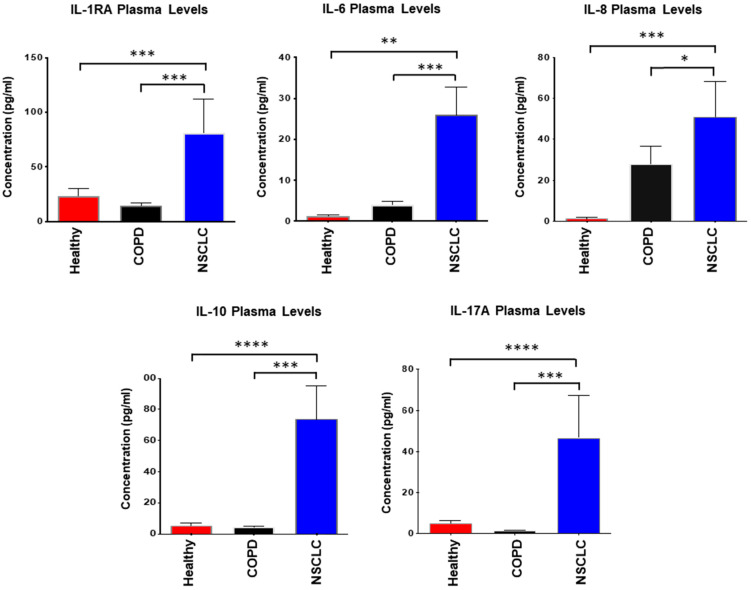
Plasma concentrations of IL-1RA, IL-6, IL-8, IL-10, and IL-17A in healthy individuals, NSCLC patients, and COPD patients. Plasma cytokine levels were quantified using a validated human multiplex ELISA platform (33-analyte panel; Eve Technologies, Calgary, AB, Canada). Data are presented as mean values ± SEM. Statistical comparisons between groups were performed using the t-test with Welch’s correction for unequal variances. Asterisks indicate significant differences in analyte levels in NSCLC patients relative to healthy individuals or COPD patients: *p* ≤ 0.05 (*), *p* < 0.03 (**), *p* < 0.004 (***), *p* < 0.0002 (****).

**Figure 3 ijms-26-11014-f003:**
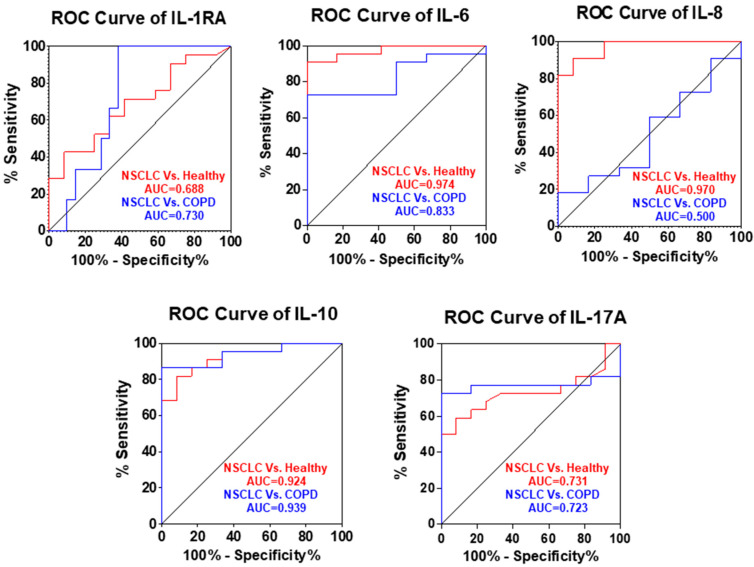
ROC curves illustrating the diagnostic accuracy of individual plasma biomarkers (IL-1RA, IL-6, IL-8, IL-10, and IL-17A) for lung cancer detection. Each curve represents the performance of a single cytokine, with the area under the curve (AUC), indicating its diagnostic value. IL-6, IL-8, and IL-10 showed the highest AUC values, suggesting their superior performance as blood-based biomarkers.

**Figure 4 ijms-26-11014-f004:**
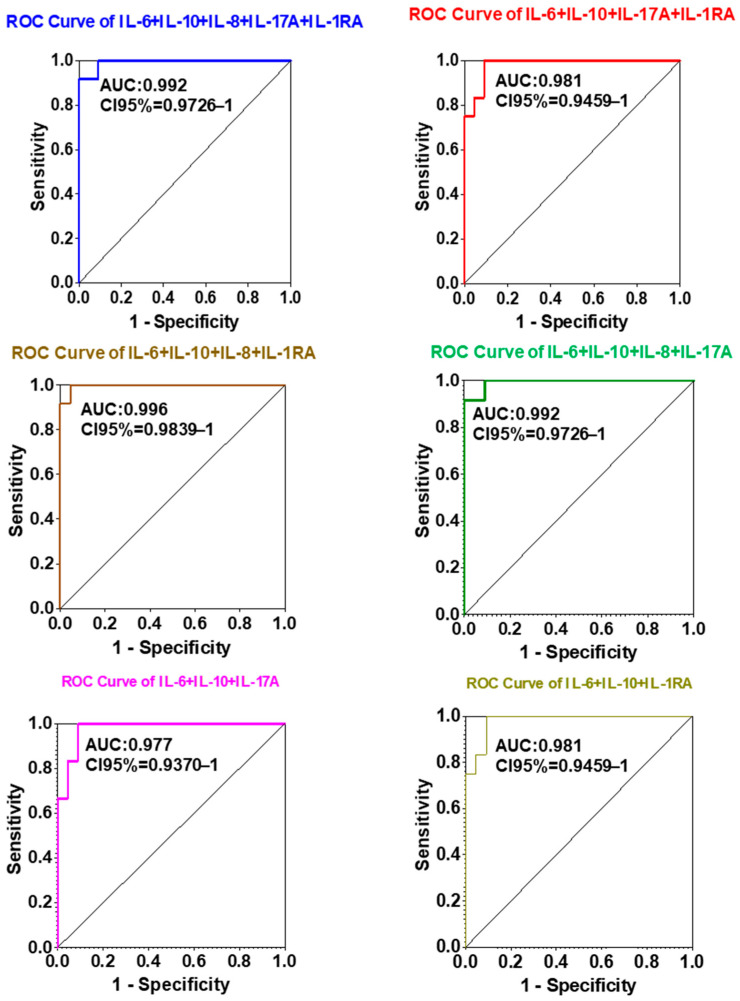
ROC curves illustrating the diagnostic performance (specificity vs. sensitivity) of various biomarker combinations for lung cancer detection. Each curve represents a distinct multiplex panel, with the area under the curve (AUC) indicating the accuracy of each combination. The IL-6 + IL-10 + IL-8 + IL-1RA panel achieved the highest diagnostic AUC (0.996), demonstrating the potential of multiplex biomarker analysis for highly accurate diagnosis.

**Table 1 ijms-26-11014-t001:** Clinico-pathological characteristics of the selected population.

Characteristics	NSCLC (n = 22)	COPD (n = 6)	Normal (n = 12)
Age (years)	72 (61, 84)	64 (62, 67)	63 (60, 69)
Gender			
Male	11	3	8
Female	11	3	4
Pathological types			
Squamous cell carcinoma	11	-	-
Adenocarcinoma	11	-	-
Clinical stage			
Stage IA	11	-	-
Stage IB	7	-	-
Stage IIA	4	-	-

**Table 2 ijms-26-11014-t002:** Plasma levels (pg/mL) of IL-1RA, IL-6, IL-8, IL-10, and IL-17A in patients with different NSCLC pathological types. *p*-value (squamous cell carcinoma vs. adenocarcinoma).

Pathological Types	n	IL-1RA	IL-6	IL-8	IL-10	IL-17A
Squamous cell carcinoma	11	38.35 ± 10.41	31.75 ± 12.20	52.87 ± 23.06	65.71 ± 30.31	64.91 ± 40.72
Adenocarcinoma	11	128.00 ± 62.04	20.14 ± 6.17	49.08 ± 26.73	86.66 ± 31.02	28.32 ± 9.49
*p*-value		0.20	0.41	0.92	0.72	0.40

## Data Availability

The original contributions presented in this study are included in the article. Further inquiries can be directed to the corresponding author(s).

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
