# Peer review of "Hematological Biomarkers for Early Detection of Lung Cancer: Evaluating the Diagnostic Potential of Circulating Interleukin Levels"

_ijms, 2025, doi:10.3390/ijms262211014_

Round 1
Reviewer 1 Report
Comments and Suggestions for Authors
Nice work,congrats!
Few points of major impact you should take under consideration: 1)add the WHO classification for thoracic tumors,2)it is very important to know if each case had or not and in what grade or type of inflammation in the microenvirement of the tumor.Squamous and adeno is not so important(you found out allready).3)you sure that the PDX were immunocompetent?4)PD-L1 and CTLA4 status?5)Prexisting therapies that can alterate the results?
Author Response
"Please see the attachment."

Reviewer 2 Report
Comments and Suggestions for Authors
- For Figure 1, it’s recommended that the authors place the bars from the two groups side by side with statistical annotations, rather than stacking them in one column.
- The variation is so huge in Figure 1. How do the authors explain this?
- For improved readability, the authors should directly place the statistical significance indicators in their corresponding positions.
- What’s the difference in these interleukins between patients with different tumor stages?
- The layout of Figure 3 is too sparse. The authors should rearrange the curve images in Figure 3 and add the AUC value into each curve.
- Since numerous benign diseases can also lead to elevated interleukin levels, the present results cannot prove the significance of IL levels in lung cancer screening. The authors should include more patients with non-tumor lung diseases to better demonstrate the specificity of interleukins for lung cancer.
Author Response
"Please see the attachment."

Reviewer 3 Report
Comments and Suggestions for Authors
The study evaluates plasma interleukins as blood-based biomarkers for early detection of non-small cell lung cancer (NSCLC), including comparison with COPD and healthy controls, and explores diagnostic performance of single cytokines and multiplex panels. The topic is clinically relevant and potentially translational. The design combines a preclinical xenograft screen with a human case-control cohort, examining multiple cytokines and ROC-based performance. The overall framework is sound and the data preliminarily support the feasibility of interleukin profiling for NSCLC detection. However, several spects require strengthening. The detailed issues are as follows:
The p-value in Figure 2: a *p is missing and an extra ***p is present.
The formatting of "IL6," "IL8," and "IL10" in the subsection headings is inconsistent with the "IL-6," "IL-8," and "IL-10" used in the main text. Besides, IL-1RA and IL-1A
Author Response
"Please see the attachment."

Round 2
Reviewer 1 Report
Comments and Suggestions for Authors
OK,Thank you
Author Response
"Please see the attachment."

Reviewer 2 Report
Comments and Suggestions for Authors
As all the clinical parts of this study were based on less than 50 cases, the authors should emphasize the limitation and the potential statistical bias due to the small sample size.
It’s still strongly recommended that the authors collect more cases to make their findings and conclusions more robust and reliable.
Author Response
"Please see the attachment."

Round 3
Reviewer 2 Report
Comments and Suggestions for Authors
The authors have made the corresponding revisions. Publication is recommended.